# Educational level and alcohol use in adolescence and early adulthood—The role of social causation and health-related selection—The TRAILS Study

Heiko Schmengler[1]*, Margot Peeters[1], Anton E. Kunst[2], Albertine J. Oldehinkel[3], Wilma A. M. Vollebergh[1]

1 Department of Interdisciplinary Social Science, Utrecht Centre for Child and Adolescent Studies, Utrecht University, Utrecht, the Netherlands, 2 Department of Public and Occupational Health, Amsterdam UMC, University of Amsterdam, Amsterdam, the Netherlands, 3 Department of Psychiatry, Interdisciplinary Center Psychopathology and Emotion Regulation, University Medical Center of Groningen, University of Groningen, Groningen, the Netherlands

* h.schmengler@uu.nl

**Data Availability Statement:** Under the General Data Protection Regulation (GDPR), our dataset is considered pseudonymized rather than

## Abstract

Both social causation and health-related selection may influence educational gradients in alcohol use in adolescence and young adulthood. The social causation theory implies that the social environment (e.g. at school) influences adolescents' drinking behaviour. Conversely, the health-related selection hypothesis posits that alcohol use (along other health-related characteristics) predicts lower educational attainment. From past studies it is unclear which of these mechanisms predominates, as drinking may be both a cause and consequence of low educational attainment. Furthermore, educational gradients in alcohol use may reflect the impact of 'third variables' already present in childhood, such as parental socioeconomic status (SES), effortful control, and IQ. We investigated social causation and health-related selection in the development of educational gradients in alcohol use from adolescence to young adulthood in a selective educational system. We used data from a Dutch population-based cohort (TRAILS Study; n = 2,229), including measurements of educational level and drinking at ages around 14, 16, 19, 22, and 26 years (waves 2 to 6). First, we evaluated the directionality in longitudinal associations between education and drinking with cross-lagged panel models, with and without adjusting for pre-existing individual differences using fixed effects. Second, we assessed the role of childhood characteristics around age 11 (wave 1), i.e. IQ, effortful control, and parental SES, both as confounders in these associations, and as predictors of educational level and drinking around age 14 (wave 2). In fixed effects models, lower education around age 14 predicted increases in drinking around 16. From age 19 onward, we found a tendency towards opposite associations, with higher education predicting increases in alcohol use. Alcohol use was not associated with subsequent changes in education. Childhood characteristics strongly predicted education around age 14 and, to a lesser extent, early drinking. We mainly found evidence for the social causation theory in early adolescence, when lower education predicted increases in subsequent alcohol use. We found no evidence in support of the health-related selection

**Funding:** This study is made possible by the Consortium on Individual Development (CID). CID is funded through the Gravitation program of the Dutch Ministry of Education, Culture, and Science and the Netherlands Organization for Scientific Research (NWO) (grant number 024.001.003). This research is part of the TRacking Adolescents' Individual Lives Survey (TRAILS). Organizations participating in TRAILS include various departments of the University Medical Center and University of Groningen, the Erasmus University Medical Center Rotterdam, Utrecht University, the Radboud Medical Center Nijmegen, and the Parnassia Bavo group, all in the Netherlands. TRAILS has been financially supported by various grants from NWO, ZonMW, GB-MaGW, the Dutch Ministry of Justice, the European Science Foundation, BBMRI-NL, and the participating universities. The funders had no role in study design, data collection and analysis, decision to publish, or preparation of the manuscript.

**Competing interests:** The authors have declared that no competing interests exist.

hypothesis with respect to alcohol use. By determining initial educational level, childhood characteristics also predict subsequent trajectories in alcohol use.

## Introduction

Lower socioeconomic status (SES) has been associated with increased alcohol-related morbidity in adulthood [1, 2]. To understand the mechanisms by which SES and alcohol-related outcomes become associated, it is important to focus on adolescence and young adulthood, as this is when alcohol use is initiated, and youngsters can affect their own later SES through education [3]. Indeed, many studies have shown that alcohol use is associated with lower adolescent educational attainment in the selective educational systems common in Western Europe. These educational systems are characterized by an early selection into different classrooms–and hereby different social contexts–based on academic aptitude. Subsequently, a proportion of students is mobile mostly between adjacent educational tracks [4]. In a Dutch sample of 12 to 16-year-olds, the prevalence of past month alcohol use was 31.1% in the lower vocational track, whilst it was only 12.2% in the academic track [5]. Similar results have been found in other countries with selective educational systems, such as Belgium, Austria, and Germany [6, 7]. However, the mechanisms by which educational trajectories and drinking behaviours become associated remain poorly understood [8, 9].

Two mechanisms may explain educational gradients in alcohol use: social causation and health-related selection [8–10]. The social causation theory implies that the social environment (e.g. at school) predicts adolescents' drinking behaviour [8]. For example, educational tracks may differ in terms of future expectations and alcohol-related norms. Students in the lower tracks may more frequently experience feelings of futility, have poor future prospects and low self-esteem, and hence may turn to risk behaviours as means to gain recognition amongst peers [7, 10, 11]. Conversely, the health-related selection hypothesis posits that poor health behaviours, such as an early onset of drinking, predict lower academic achievement and may lead to downward or impair upward social mobility in the educational system [8]. For example, early onset heavy alcohol use has been associated with cognitive impairments in adolescents, and may hereby negatively affect performance at school [12, 13]. Importantly, social causation and health-related selection are not mutually exclusive, may reinforce each other over time, but differ in their relative importance in different phases of adolescence.

In addition, associations between educational level and alcohol use may reflect the impact of 'third variables' (i.e. confounders) already present in childhood [8, 14]. On the individual level, differences in psychological dispositions may impact both initial selection into educational tracks and later substance use [8]. For example, students with high levels of effortful control are more likely to succeed in educational settings [15] and may also be more likely to abstain from substance use [16]. Similarly, good cognitive functioning is related to both higher educational attainment and better health behaviours later in life [17, 18]. On the social (environmental) level, characteristics of the family environment may influence both educational prospects and alcohol use. Adolescents from lower SES families may more frequently be exposed to harmful drinking at home [2, 19], and lower SES families may also have less resources to support the education of their children [4]. These mechanisms may be referred to as 'indirect' social causation or health-related selection, depending on whether emphasis is given to the social-environmental or individual-level factors [8–10]. Like the 'direct' mechanisms, the 'indirect' mechanisms can work in conjunction. For example, higher SES parents may be able to provide a safer and more stimulating family environment, positively

influencing their young child's emotional and cognitive development [20, 21]. Resulting differences in psychological dispositions can then predict both later educational attainment and health behaviours.

Only few studies have investigated the temporal directions of associations between alcohol use and education throughout adolescence [22–24]. One method to study these is by modelling autoregressive and cross-lagged associations simultaneously in cross-lagged panel models (CLPMs). Overall, results were mixed and do not provide conclusive evidence for the dominance of either the 'direct' social causation or the 'direct' health-related selection hypothesis. In the USA, Crosnoe mainly found evidence for social causation, as academic failure predicted more subsequent drinking in 16-year-old adolescents, but not vice versa [22]. Conversely, Latvala et al. mainly found evidence for health-related selection in Finnish early and mid-adolescents [24]. Higher alcohol use around age 12 predicted lower GPA around age 14, and higher alcohol use around age 14 was associated with a lower likelihood of being in education around age 16. Latvala et al. found no evidence for social causation effects throughout adolescence, except for one significant path from higher education around age 17 to increased alcohol use around age 24 [24]. Similar to Latvala and colleagues, Owens et al. found health-related selection effects, with alcohol use predicting lower GPA one year later from around age 14 through around age 18 in a US sample. Social causation results were less consistent and revealed mixed findings, with higher GPA predicting less alcohol use in younger and more alcohol use in older adolescents [23].

This heterogeneity in results may partially relate to the fact that CLPMs cannot separate the within and between-person variances of the cross-lagged variables. Therefore, they are unable to rule out the possibility of confounding by unmeasured time-invariant (or trait-like) 'third variables' often already present prior to adolescence [25]. These variables may include relatively stable genetic or temperamental factors, which are both associated with health behaviours and educational outcomes, as mentioned above [8]. Unmeasured heterogeneity affecting reciprocal associations can be addressed with novel statistical methods, which combine CLPMs with fixed effects methods, allowing to assess bidirectional associations between education and alcohol use at the within-person level [26].

In addition, all past studies were from comprehensive educational systems and results may not be applicable to the selective educational systems common in Western Europe. In the Netherlands, this selection into one of four tracks (Fig 1) takes place at age 11–12, based on a large battery of cognitive tests and the advice of the teacher in primary education [4]. The Dutch system may allow assessing social mobility in adolescents at an earlier age than possible in comprehensive educational systems, as social stratification already occurs around the beginning of adolescence.

In the current study, we aimed to contribute to the literature by addressing the following research question: "To what extent do the social causation and health-related selection hypotheses explain educational differences in alcohol use in adolescents and young adults in a selective educational system?" To answer our research question, we modelled reciprocal relationships (i.e. 'direct' social causation and health-related selection) between alcohol use and educational level throughout adolescence and young adulthood. Furthermore, we aimed to evaluate the role of family SES, as well as childhood effortful control and IQ, which may all act both as determinants of the initial selection into educational tracks and early alcohol use, and as confounders ('third variables') in the cross-lagged paths. Finally, we addressed potential residual confounding by unmeasured relatively stable 'third variables' in bidirectional associations, using a fixed effects approach.

Based on past findings highlighting the strong role of the social environment in adolescent drinking [7, 10, 11], as well as the deleterious effect of intensive alcohol use on cognitive

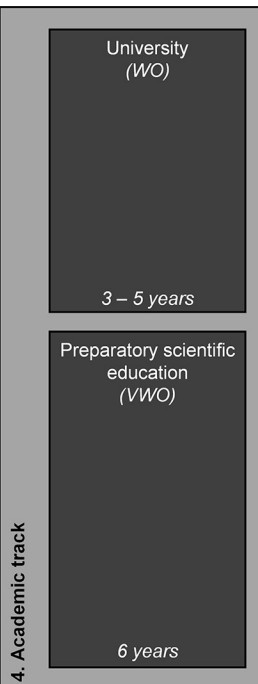
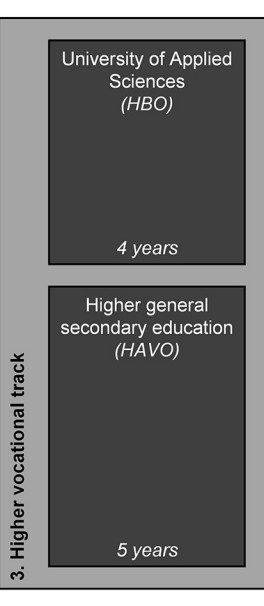
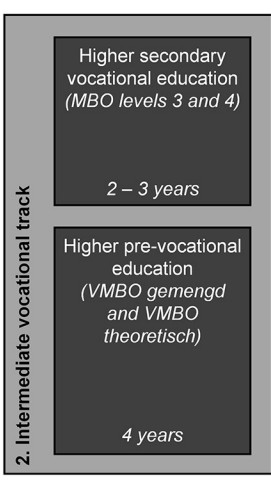
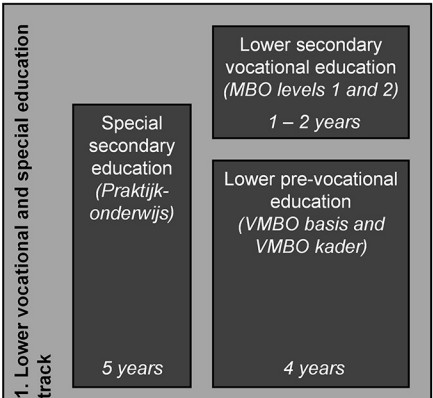
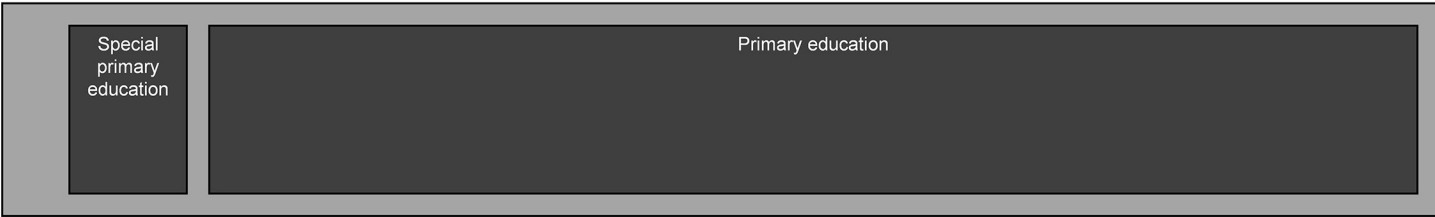

**Fig 1. The Dutch educational system.**

performance [12, 13], we expected to find evidence for both mechanisms in CLPMs (i.e. lower education predicting increases in alcohol use and vice versa). We also expected an attenuation in cross-lagged associations after adjusting for childhood characteristics, in line with the 'indirect' social causation and health-related selection hypotheses [8]. During the transition from childhood to early adolescence, we expected to find associations between childhood psychological characteristics (i.e. IQ and effortful control) and the educational track in which adolescents were initially placed, in line with the health-related selection hypothesis. Finally, we expected to find associations between higher parental SES and higher educational level, as well as lower alcohol use, in early adolescence, in line with the social causation theory.

## Materials and methods

### Study population

We used data from the first six waves (T1 –T6) of the TRacking Adolescents' Individual Lives Survey (TRAILS), a population-based prospective cohort study of Dutch adolescents. A detailed description of the cohort can be obtained elsewhere [27]. At the beginning of the study, 135 schools in the province of Groningen were invited, of which 122 decided to participate [28]. Adolescents were followed between 2000 and 2017 with assessments around age 11, 14, 16, 19, 22, and 26. Ethical approval for TRAILS was obtained from the Dutch national

ethics committee Central Committee on Research Involving Human Subjects (#NL38237.042.11). Written informed consent was obtained from both adolescents and their parents prior to inclusion in the study.

## Alcohol use

Alcohol use was assessed contemporaneously from waves 2 through 5 by self-report using a quantity-frequency measure [29]. Quantity-frequency measures of alcohol use have shown adequate/good validity and reliability across studies [30]. Frequency was assessed by asking adolescents about the number of weekdays (Monday to Thursday) and weekend days (Friday to Sunday) on which alcohol was consumed. Quantity was measured by asking about the average number of alcoholic beverages consumed on a typical week or weekend day (9-point scale ranging from 1 = 'I never drink on a weekday/ weekend day' to '11 glasses or more'). A quantity-frequency measure was obtained by multiplying the quantity scores for week and weekend days by the corresponding frequency scores, and then summing both scores [29]. At wave 6, alcohol use was assessed by self-report using the sum score of the AUDIT-C, which consists of the first three items of the Alcohol Use Disorders Identification Test (AUDIT) [31]. The AUDIT-C (Cronbach's alpha = 0.66) includes each one item assessing frequency (number of drinking occasions in past 12 months; range: never to $\geq$ 4 per week), quantity (number of glasses per typical drinking occasion; range: 1–2 to $\geq$ 10), as well as a measure of binge drinking (number of occasions where $\geq$ 6 glasses of alcohol are consumed; range: never to daily or almost daily). Both measures were z-score transformed before inclusion in our analyses. It was not possible to compute alcohol use scores for a number of participants who had missing information for at least either quantity or frequency items (waves 2–5), or at least one item of the AUDIT-C (wave 6): wave 2: N = 190, 8.85%; wave 3: N = 249, 13.70%; wave 4: N = 263, 13.99%; wave 5: N = 315, 17.69%; wave 6: N = 438; 27.10%.

## Adolescents' educational level

The Dutch educational system is characterized by an early (age 11–12) selection into a particular educational track, based on a battery of cognitive tests and the advice of the primary school. In line with this selection, we used a measure of educational level that is consistent throughout all of secondary and tertiary education (Fig 1). There are four tracks in the Dutch educational system, each consisting of a specific type of secondary school followed by tertiary education at the corresponding level: 1. lower vocational track, 2. intermediate vocational track, 3. higher vocational track, 4. academic track. In addition, there is a special education track, attended by students who are unable to attend regular education. This track was collapsed with the lower vocational track. Educational track membership was assessed from wave 2 to 6 by asking for participants' current enrolment, as well as their highest completed diploma. Participants who finished the final diploma of a given track received the value corresponding to that level for all subsequent waves, unless they continued education at a higher level. Educational level was not assessed at wave 1, since most children were still in elementary school. Our measure of educational level allows us to assign a score that represents an age-appropriate measure of educational attainment as proxy of developing SES.

Missing information on educational track membership from waves 2 through 6 was filled in using retrospective event history calendars conducted at wave 3 and wave 5. Participants who were still in elementary education or in a combined class at wave 2 were assigned according to their elementary school teachers' recommended level. If this information was not available, pupils were classified according to the first track they attended after leaving elementary education or the combined class. It was not possible to classify a number of participants, who

were not in education for a longer period, were not classifiable into an educational track (e.g. because of education abroad), whose educational level was assessed incompletely, who did not respond to questions on education, or who had permanently left the educational system (wave 2: N = 221, 10.29%; wave 3: N = 289, 15.90%; wave 4: N = 373, 19.84%; wave 5: N = 352, 19.76%; wave 6 = 424, 26.24%). Educational level was set to missing for these participants.

## Characteristics at baseline (wave 1)

Characteristics at baseline hypothesized to be associated with both alcohol use and education were selected based on earlier studies [3, 14] and include:

1. *Parents' socioeconomic position (SES)*, constructed as the mean score of five indicators (standardized): maternal and paternal educational attainment, maternal and paternal occupational position (according to the International Standard Classification of Occupations), and family income [32].

2. *Childhood psychological characteristics* include effortful control and the Intelligence Quotient (IQ). Effortful control was assessed using the corresponding subscale from the parent-report Early Adolescent Temperament Questionnaire (EATQ-R), which consists of 11 items with 5 response categories (Cronbach's alpha = 0.86; McDonald's omega = 0.87 [33, 34]). Children's IQ was estimated using the Block Design and Vocabulary subtests of the Revised Wechsler Intelligence Scale for Children (WISC-R) [35].

3. Furthermore, we adjusted for demographic characteristics at baseline, that is, area of residence (City of Groningen, Leeuwarden, Assen, other regions), adolescent age, gender, and ethnicity. Children were classified as having non-Dutch ethnicity if at least one of their parents was born outside the Netherlands [36].

## Analytic approach

First, we computed descriptive statistics of the study population by cross-tabulating baseline characteristics (mean age 11) with early adolescent educational track membership at wave 2 (mean age 14), as well as alcohol use with concurrent educational level from wave 2 through wave 6 (mean age 26). Second, we computed cross-lagged panel models (CLPMs) between educational level and alcohol use from wave 2 through wave 6, whilst sequentially adjusting for different sets of baseline covariates. Our full covariate-adjusted CLPM included age, gender, area of residence, ethnicity, parental socioeconomic status, IQ, and effortful control at baseline (wave 1) as predictors. The CLPM estimates prospective associations between educational level and changes in subsequent alcohol use, and between alcohol use and changes in subsequent educational level, whilst taking into account temporal stability and reciprocity (S1 and S2 Figs) [26].

Third, we conducted analogous CLPMs with fixed effects. These models only use within-person variance to estimate associations between cross-lagged variables, hereby adjusting for all measured and unmeasured time-invariant characteristics. In line with the one-sided specification by Allison et al. [26], two separate fixed effects models were fit to assess lagged associations from education to changes in subsequent alcohol use, and from alcohol use to changes in subsequent education (S3 Fig). The fixed effects term was represented by a latent variable of all measurements of the outcome with each having its factor loading constrained to be 1. This latent variable was allowed to be correlated freely with all time-varying exogenous variables in the model. Reciprocal causation was accommodated by including correlations between the

error term of the outcome at each measurement occasion and all future values of the time-varying exposure.

Fourth, we evaluated the role of parental SES and adolescents' psychological characteristics at baseline (i.e. IQ, effortful control) in predicting alcohol use and educational level at wave 2 in the full covariate-adjusted CLPM. Finally, we computed intra-class correlations (ICC) for education and alcohol use over time, to assess which proportions of the variance were at the within-person and at the between-person level.

Attrition analyses showed that at wave 2 3.63% (N = 81) of the original participants no longer participated in the study. At wave 3 this was the case for 18.44% (N = 411), at wave 4 for 15.66% (N = 349), at wave 5 for 20.10% (N = 448), and at wave 6 for 27.50 (N = 613) of the original participants. Adolescents with male gender, non-Dutch ethnicity, lower educational level, IQ, and effortful control, as well as those from lower SES households were more likely to drop out of the study (S1 Table). Higher alcohol use was also related to dropout, but only significantly at wave 3. Similar differences were found when comparing participants with complete information on educational level to those whose educational level was missing or could not be classified (S2 Table). To deal with missing information, full-information maximum likelihood (FIML) was implemented, allowing to incorporate information from all participants. All variables that were not part of each respective analytic model were included as auxiliary variables to adjust for potential bias due to missing data on these variables [37]. Model fit in Structural Equation Models (SEM) was assessed using the Comparative Fit Index (CFI), the Tucker Lewis Index (TLI), the Root Mean Square Error of Approximation (RMSEA), and the Standardised Root Mean Residual (SRMR). Following the suggestions by Hu & Bentler, model fit was judged as 'good' if the CFI and TLI were >0.95, the RMSEA was <0.06, and the SRMR was <0.08 [38]. Standard errors were estimated using robust maximum likelihood (MLR) to take into account potential normality violations. Analyses were conducted in Mplus 8.6.

## Sensitivity analyses

To determine whether the fact that a different indicator of alcohol use was used at wave 6 (i.e. the AUDIT-C) influenced our results, we conducted a sensitivity analysis in which the binge drinking item was removed from the AUDIT-C, creating a quantity-frequency measure that is somewhat similar to the one used from wave 2 to 5. Additionally, we assessed whether the ordinal nature of our educational variable affected the linear regression results by executing the bivariate CLPM and the fixed effects models using the Bayes estimator in Mplus, whilst declaring all endogenous measurements of educational level as "categorical". Overall, the results of the sensitivity analyses were similar to our main results.

## Results

### Descriptive statistics

Table 1 shows the characteristics of TRAILS participants around age 11 according to educational level around age 14. Children with less affluent or non-Dutch parents more commonly attended the lower educational tracks. Girls more frequently attended the academic and intermediate vocational tracks than boys. Children in the lower vocational track and the academic track were slightly older at baseline than those in the intermediate and higher vocational tracks. Further, higher IQ and higher effortful control around age 11 predicted higher education around age 14.

Table 2 shows educational level around age 14, 16, 19, 22, and 26, and concurrent alcohol use. Around age 14 and 16, we found an educational gradient in alcohol use, with adolescents in the lower tracks consuming more alcohol compared to those in the higher tracks. No

**Table 1. Characteristics of adolescents participating in the TRAILS Study (the Netherlands, 2000–2017, N = 2,229) at wave 1 (2000–2002) and according to educational level at wave 2 (2003–2005).**

| | All levels | | Lower vocational & special education | | Intermediate vocational | | Higher vocational | | Academic | |
|---|---|---|---|---|---|---|---|---|---|---|
| | N = 2,229 | | N = 635 | | N = 497 | | N = 383 | | N = 457 | |
| Male gender, N (%) | 1,098 | (49.26) | 341 | (53.70) | 217 | (43.66) | 196 | (51.17) | 195 | (42.67) |
| District, N (%) | | | | | | | | | | |
| *City of Groningen* | 794 | (35.62) | 227 | (35.75) | 157 | (31.59) | 128 | (33.42) | 197 | (43.11) |
| *Leeuwarden* | 596 | (26.74) | 193 | (30.39) | 125 | (25.15) | 84 | (21.93) | 124 | (27.13) |
| *Assen* | 489 | (21.94) | 127 | (20.00) | 98 | (19.72) | 124 | (32.38) | 102 | (22.32) |
| *Other regions* | 350 | (15.70) | 88 | (13.86) | 117 | (23.54) | 47 | (12.27) | 34 | (7.44) |
| Non-Dutch ethnicity, N (%) | 301 | (13.50) | 108 | (17.01) | 61 | (12.27) | 39 | (10.18) | 45 | (9.85) |
| Age, mean (SD) | 11.11 | (0.56) | 11.16 | (0.56) | 11.07 | (0.54) | 11.05 | (0.56) | 11.14 | (0.56) |
| Parental socioeconomic status (SES), mean (SD) | -0.05 | (0.80) | -0.53 | (0.70) | -0.16 | (0.67) | 0.21 | (0.68) | 0.55 | (0.70) |
| Wechsler Intelligence Deviation Quotient (IQ), mean (SD) | 97.19 | (15.00) | 86.05 | (12.49) | 95.20 | (10.98) | 102.68 | (11.20) | 111.14 | (11.91) |
| Effortful control, mean (SD) | 3.23 | (0.68) | 2.92 | (0.62) | 3.06 | (0.63) | 3.35 | (0.65) | 3.65 | (0.61) |

SD = standard deviation.

**Table 2. Alcohol use of adolescents and young adults participating in the TRAILS Study (the Netherlands, 2000–2017, N = 2,229) according to concurrent educational level.**

| | Wave 2 | | Wave 3 | | Wave 4 | | Wave 5 | | Wave 6 | |
|---|---|---|---|---|---|---|---|---|---|---|
| | N = 2,148 | | N = 1,818 | | N = 1,880 | | N = 1,781 | | N = 1,616 | |
| Date range | 2003–2005 | | 2005–2008 | | 2008–2010 | | 2012–2014 | | 2016–2017 | |
| Age, mean (SD) | 13.57 | (0.53) | 16.28 | (0.71) | 19.08 | (0.60) | 22.29 | (0.65) | 25.66 | (0.60) |
| Male gender, N (%) | 1,054 | (49.07) | 867 | (47.69) | 898 | (47.77) | 843 | (47.33) | 735 | (45.48) |
| Educational level, N (%) | | | | | | | | | | |
| *Lower vocational & special education* | 635 | (32.20) | 349 | (22.83) | 161 | (10.68) | 136 | (9.52) | 78 | (6.54) |
| *Intermediate vocational* | 497 | (25.20) | 405 | (26.49) | 498 | (33.02) | 354 | (24.77) | 273 | (22.90) |
| *Higher vocational* | 383 | (19.42) | 362 | (23.68) | 475 | (31.50) | 594 | (41.57) | 489 | (41.02) |
| *Academic* | 457 | (23.17) | 413 | (27.01) | 374 | (24.80) | 345 | (24.14) | 352 | (29.53) |
| Alcohol use | | | | | | | | | | |
| Quantity-frequency score, mean (SD) | | | | | | | | | | |
| *All levels* | 1.64 | (4.56) | 6.95 | (9.56) | 10.18 | (11.64) | 10.18 | (11.01) | - | - |
| *Lower vocational & special education* | 2.20 | (6.12) | 9.70 | (13.52) | 11.12 | (15.48) | 11.35 | (15.78) | - | - |
| *Intermediate vocational* | 1.76 | (3.73) | 6.74 | (8.29) | 9.47 | (9.97) | 8.56 | (8.96) | - | - |
| *Higher vocational* | 1.58 | (4.14) | 5.87 | (8.81) | 10.61 | (11.50) | 10.70 | (11.08) | - | - |
| *Academic* | 0.84 | (3.45) | 4.69 | (5.05) | 9.46 | (10.46) | 10.69 | (10.98) | - | - |
| AUDIT-C score, mean (SD) | | | | | | | | | | |
| *All levels* | - | - | - | - | - | - | - | - | 4.60 | (2.41) |
| *Lower vocational & special education* | - | - | - | - | - | - | - | - | 3.81 | (2.42) |
| *Intermediate vocational* | - | - | - | - | - | - | - | - | 4.23 | (2.43) |
| *Higher vocational* | - | - | - | - | - | - | - | - | 4.55 | (2.33) |
| *Academic* | - | - | - | - | - | - | - | - | 4.92 | (2.30) |

SD = standard deviation.

general gradient was seen around age 19, and around age 21 young adults in the intermediate vocational track consumed less alcohol than those in the other three tracks. By age 26, the educational gradient in alcohol use was reverse, with young adults in the higher tracks scoring higher on the AUDIT-C.

### Cross-lagged associations between educational level and alcohol use

In Fig 2, we assessed the 'direct' social causation and health-related selection hypotheses by evaluating bidirectional associations between educational level and alcohol use from age 14 to 26, using CLPMs. In bivariate CLPMs (Fig 2, Model 1), educational level exhibited very high stability (ß > 0.80), while the stability of alcohol use was lower and increased over time, ranging from approximately 0.20 in early adolescence to 0.60 in young adulthood. The ICC for education was 0.820, which indicates that 82% of the variance across the five measurements of educational level was due to differences between persons. This finding is in line with the high stability in educational level we found in CLPMs. The ICC for alcohol use was 0.293, suggesting that a substantial proportion of variance in alcohol use represents within-person fluctuations over time.

When considering social causation paths from educational level to alcohol use, lower education around age 14 predicted increases in alcohol use around age 16 (ß coefficient -0.140, Standard Error [SE] 0.022, p<0.001). Conversely, from age 16 onwards we found consistent, though relatively weaker, associations between higher education and increases in subsequent alcohol use (from around age 16 to 19: ß 0.069, SE 0.023, p = 0.003; from around age 19 to 22: ß 0.069, SE 0.026, p = 0.007; from around age 22 to 26: ß 0.069, SE 0.026, p = 0.008). In multivariate CLPMs, adjustment for parental SES led to the greatest change in the model (S4 Fig, Model 3), rendering associations between education around 16 and increases in alcohol use around 19 (ß 0.033, SE 0.026, p = 0.198), as well as between education around 22 and increases

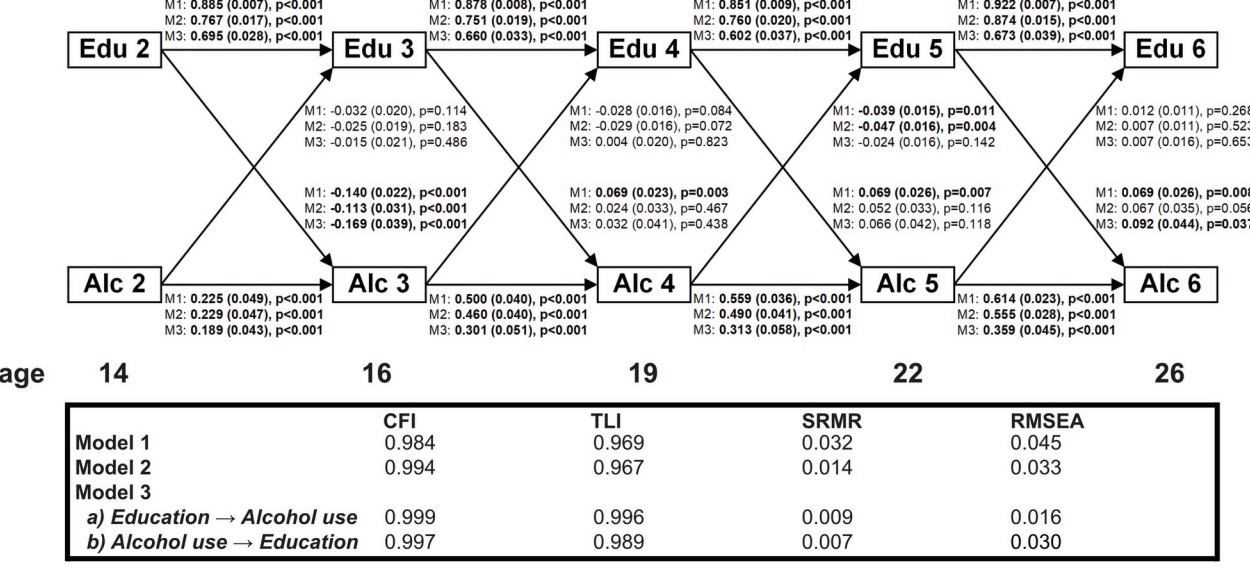

**Fig 2. Bidirectional associations between educational level and alcohol use in the TRAILS Study (the Netherlands, 2000–2017, N = 2,229); linear regression coefficients (stdyx-standardized ß-coefficient, robust standard error, p-value) from cross-lagged panel models without (Model 1 and 2) and with fixed effects (Model 3).** Model 1: bivariate cross-lagged panel model. Model 2: cross-lagged panel model adjusted for age, gender, area of residence, ethnicity, parental socioeconomic status, IQ, and effortful control at baseline (wave 1). Model 3: cross-lagged panel models with fixed effects–adjustment for time-invariant characteristics was performed by inclusion of a latent variable. Edu = educational level; Alc = alcohol use. **Boldface** denotes statistical significance at p < 0.05.

in alcohol use around 26 (ß 0.035, SE 0.029, p = 0.233) insignificant. In the full covariate-adjusted model (Fig 2, Model 2), only the association between lower education around age 14 and increases in alcohol use around age 16 remained statistically significant (ß -0.113, SE 0.031, p<0.001). Results from the fixed effects model (Fig 2, Model 3a) somewhat resembled the full covariate-adjusted model (Fig 2, Model 2). Lower education around age 14 significantly predicted increases in alcohol use around 16 (ß -0.169, SE 0.039, p<0.001). From age 19 to 26 we found tendencies towards opposite associations, though only the association between higher education around age 22 and increases in alcohol use around age 26 was once more significant in the fixed effects model (ß 0.092, SE 0.044, p = 0.037). However, in our sensitivity analysis (S5 Fig) this association failed to reach significance both in the fixed effects model (ß 0.073, SE 0.045, p = 0.106) and in the bivariate CLPM (ß 0.044, SE 0.028, p = 0.114) once the binge drinking item was removed from the AUDIT-C. Besides this, our sensitivity analysis yielded very similar results to the main analysis.

When considering health-related selection related to alcohol use in the bivariate CLPM (Fig 2, Model 1), we found a weak prospective association between higher alcohol use and decreases in subsequent educational level from around age 19 to 22 (ß -0.039, SE 0.015, p = 0.011). This result was robust to adjustment for covariates (ß -0.047, SE 0.016, p = 0.004) (Fig 2, Model 2), but did not survive in the fixed effects model (ß -0.024, SE 0.016, p = 0.142) (Fig 2, Model 3b). In our sensitivity analysis with the Bayes estimator (S6 Fig), in which educational level was declared as 'categorical', we additionally found a significant path from higher alcohol use around age 14 to decreases in education around age 16 in the bivariate CLPM (ß -0.039, posterior SD 0.014, one-tailed p = 0.002). However, similar to our main result, this effect was absent in the fixed effects model (ß -0.011, posterior SD 0.016, one-tailed p = 0.255). Model fit of all cross-lagged models was good [38].

## Associations of childhood predictors with educational level and alcohol use in early adolescence

In Table 3, we evaluated the role of childhood (around age 11) psychological characteristics (i.e. health-related selection) and parental SES (i.e. social causation) as predictors of initial

**Table 3. The association between baseline characteristics (wave 1) and educational level and alcohol use at wave 2 in the TRAILS Study (the Netherlands, 2000–2017, N = 2,229) in the multivariate-adjusted cross-lagged panel model (model 2) in Fig 2; linear regression coefficients (stdyx-standardized ß-coefficient, robust standard error, p-value); all predictors are mutually adjusted.**

|  | Educational level | Alcohol quantity-frequency score |
|---|---|---|
| **Male gender** | -0.029 (0.015), p = 0.060 | -0.024 (0.022), p = 0.267 |
| **District** |  |  |
| *City of Groningen* | ref | ref |
| *Leeuwarden* | **-0.037 (0.017), p = 0.034** | 0.004 (0.027), p = 0.896 |
| *Assen* | **-0.039 (0.019), p = 0.046** | **-0.052 (0.027), p = 0.049** |
| *Other regions* | **-0.057 (0.019), p = 0.003** | -0.008 (0.029), p = 0.767 |
| **Non-Dutch ethnicity** | 0.009 (0.016), p = 0.570 | 0.000 (0.030), p = 0.995 |
| **Age** | 0.001 (0.018), p = 0.949 | 0.002 (0.029), p = 0.956 |
| **Parental socioeconomic status (SES)** | **0.280 (0.017), p<0.001** | **-0.086 (0.021), p<0.001** |
| **Wechsler Intelligence Deviation Quotient (IQ)** | **0.462 (0.016), p<0.001** | **-0.052 (0.026), p = 0.045** |
| **Effortful control** | **0.249 (0.017), p<0.001** | -0.042 (0.024), p = 0.079 |

All predictors are mutually adjusted.

**Boldface** denotes statistical significance at p < 0.05.

educational level (around age 14) following adolescents' selection into educational tracks, in the full covariate-adjusted CLPM shown in Fig 2, Model 2. Furthermore, we evaluated the role of parental SES (i.e. social causation) and childhood psychological characteristics in early adolescent alcohol use (around age 14). The associations of the childhood predictors with educational level and alcohol use from wave 3 to 6 are shown in S3 Table.

Higher parental SES (ß 0.280, SE 0.017, p<0.001), IQ (ß 0.462, SE 0.016, p<0.001), and effortful control (ß 0.249, SE 0.017, p<0.001) significantly predicted higher educational level around age 14. Higher parental SES (ß -0.086, SE 0.021, p<0.001) and IQ (ß -0.052, SE 0.026, p = 0.045), but not effortful control (ß -0.042, SE 0.024, p = 0.079), also independently predicted lower alcohol use around age 14, although to a lesser extent. In addition, we conducted a post-hoc analysis to assess whether associations of baseline IQ and parental SES with alcohol use around age 14 would remain significant after further adjusting for adolescents' concurrent educational level (**S4 Table**). While the association of parental SES with drinking was not impacted (ß -0.079, SE 0.024, p = 0.001), baseline IQ was no longer significantly associated with drinking around age 14 in this model (ß -0.041, SE 0.034, p = 0.222).

## Discussion

Investigating bidirectional associations between alcohol use and educational level in adolescence and young adulthood, we mainly found evidence in favour of the 'direct' social causation hypothesis in early adolescence, with lower educational level around age 14 strongly predicting subsequent increases in alcohol use around age 16. From age 19 onward, we found tendencies towards opposite associations, with higher education predicting increases in alcohol use. However, these associations failed to reach significance in the adjusted models, except for the path from higher education around age 22 to increases in alcohol use around age 26 in the fixed effects model when using the full AUDIT-C at wave 6. Also, a small health-related selection effect from higher alcohol use to lower subsequent education between age 19 and 22 was no longer significant after adjusting for unmeasured stable differences. The lack of significant cross-lagged associations throughout most of adolescence in adjusted models points to the importance of relatively stable 'third variables' associated with both education and alcohol use. Furthermore, educational track membership showed very high stability, suggesting that the potential for health-related selection later in adolescence may be limited in educational systems with early selection into educational tracks. It is therefore important to also consider the role of childhood predictors, which determine this initial selection. Indeed, we found that parental SES, as well as IQ and temperamental effortful control in childhood, strongly predicted children's educational level in early adolescence. Parental SES and IQ also predicted early alcohol use, although to a lesser extent.

### Strengths and limitations

Our study has several limitations. First, attrition and missing data may have affected our results. While we addressed missing data with FIML [39], more drop-out and missingness in participants with lower educational level, higher alcohol use, and less favourable psychological dispositions (S1 and S2 Tables) may still have influenced our results. Studies suggest that low IQ, effortful control, and parental SES are important predictors of adverse outcomes in young adulthood [40]. Future research on at-risk groups is therefore essential. Second, TRAILS used a different indicator for alcohol use at wave 6 (i.e. the AUDIT-C) compared to the other waves, which additionally includes an item assessing binge drinking. However, similar results were found when the binge drinking item was removed from the AUDIT-C (S5 Fig). Third, our models do not address time-varying confounding. Certain characteristics that change

substantially throughout adolescence, such as delinquency, may both be causally related to educational level and alcohol use, and may confound cross-lagged associations [41]. Future research could investigate the contribution of such influences in relation to health-related selection and social causation effects. Fourth, by design, our study may not have captured the whole range of effects of alcohol use on educational attainment. For example, alcohol use may still have adversely affected adolescents' GPA [24, 42], but these effects may not have been consequential enough to lead to a decline in educational track. Likewise, it may still be possible that adolescents experience downward educational mobility in case of severe alcohol-related problems, e.g. repeated hospitalization with acute intoxication [43], or polysubstance use [44]. Our quantity-frequency measure might not have adequately captured severe forms of alcohol use or the use of alcohol in combination with other substances (i.e. polysubstance use). Further research on the educational consequences of severe alcohol-related problems and polysubstance use in adolescence is warranted. Fifth, the social meaning of alcohol use might change over time, which could lead to longitudinal measurement non-invariance. However, recent research suggests that alcohol use measures are quite reliable and consistent in measuring alcohol use during adolescence and young adulthood [39, 45]. Sixth, whilst CLPMs account for the longitudinal structure of our data regarding the order of measurement occasions, they assume equal distances between waves. If the time intervals between measurement occasions are strongly unequal, CLPMs may yield biased results [46]. There were some differences in the spacing between the waves in TRAILS, which might be difficult to avoid in studies covering very long time periods (between wave 2 and 3 on average 2.71 years; between wave 3 and 4 on average 2.80 years; between wave 4 and 5 on average 3.21 years, and between wave 5 and 6 on average 3.37 years). While this might have introduced minor bias in the regression coefficients, we do not expect that this should have led to major changes in the relationships we found. Some waves started a bit later than others, and some adolescents participated in the one wave a bit later and in the following wave a bit earlier, but overall, the differences in the spacing between waves were small and on average only concerned several months.

Finally, limitations relate to the generalizability of our findings. The TRAILS participants were likely to start drinking during times when the Netherlands was at the top of international rankings of alcohol consumption amongst 12-16-year-olds. Dutch parents have since adopted more restrictive alcohol-related parenting practices [5]. Higher socioeconomic and educational groups tend to be faster at adopting behavioural innovations, which often leads to a (temporary) widening in inequalities in health behaviours [8, 47]. In the Netherlands, differences in the prevalence of early adolescents´ past month drunkenness between the lowest and the highest educational tracks have increased from 4.6% in 2003 to 9.8% in 2015 [5]. Therefore, we hypothesize that associations might be stronger in more recent Dutch cohorts than in TRAILS. Generalizability of our findings to other geographical contexts may also be limited, because both educational systems and adolescents' drinking cultures vary widely across countries [5, 48].

The long follow-up and high response rate are key strengths of our study. By using CLPMs, we take into account reciprocity between educational level and alcohol use and disentangle their temporal direction [26]. By comparing multivariate adjusted CLPMs to CLPMs with fixed effects, we adjust for both measured and unmeasured time-invariant confounding [24, 26]. We add to the literature by for the first time modelling bidirectional associations between educational level and alcohol use in a selective educational system, which provides a consistent and age-appropriate measure of educational attainment, as proxy for developing SES, over the course of adolescence. The selection into educational tracks at an age as early as 11–12 years means that Dutch adolescents grow up in distinct educational environments that are characterized by different social norms, future expectations, cognitive resources, and occupational

prospects [7, 11]–characteristics that are closely related to conceptualizations of SES in adulthood [8]. One could therefore argue that in selective educational systems, such as in the Netherlands, youngsters move into 'their own' SES at a much earlier age than in comprehensive systems, such as in Finland or the USA. Therefore, TRAILS provides a unique opportunity to investigate both the antecedents and consequences, in terms of health-related characteristics, of this differentiation and subsequent intragenerational social mobility in adolescents and young adults.

## Interpretation of findings

We found that lower educational level significantly predicted a stronger escalation in alcohol use in early adolescence only, and not later throughout adolescence and young adulthood. This result may point towards an important role for educational differences in peer group composition and social norms in early adolescence, processes which have been found to be amongst the strongest determinants of underage alcohol use [49]. Previous research has shown that schools in the lower educational tracks more commonly feature a culture characterized by feelings of futility, poor future prospects, and low self-esteem. Consequently, students may turn to alternative means to attain status amongst their peers, which may include substance use [7, 10, 11]. Indeed, early adolescents in the lower educational tracks in the Netherlands more frequently perceive substance use as 'adult-like' behaviour [50], which may be used to gain popularity with drinking peers [51]. As a result, young adolescents in these tracks may show an earlier escalation in their alcohol use.

From late adolescence onward, we found tendencies towards opposite associations, with higher educational level predicting increases in drinking. However, most of these associations did not survive statistical adjustment, and were partially explained by differences in parental SES (S4 Fig and S3 Table). This is in line with previous research, which found negative correlations between parental SES and underage drinking, and positive correlations with drinking in young adulthood [52]. Changes in social norms within educational tracks as adolescents get older could also have contributed to positive associations between higher education and increases in alcohol use in young adulthood. Adolescents moving into higher education may experience a lifting of constraints on drinking combined with a strong peer pressure towards alcohol use in the context of university culture [53, 54]. Meanwhile, adolescents who complete the vocational tracks begin fulltime employment and experience earlier transitions to adult work and family roles [39, 55, 56]. They may as a result be less likely to further escalate in their drinking. Past studies from the US and other countries show increased alcohol use amongst young adults in higher education. This 'college effect', however, may only be of temporary nature, as many young adults mature out of heavy drinking sometime after leaving university [24, 57–61].

We found no evidence of health-related selection resulting from alcohol use, as the small lagged association we found in late adolescence did not survive in the fixed effects model. Importantly, we also found no clear evidence for confounding by IQ and effortful control in the CLPMs. The selection effect we found in the bivariate CLPM may therefore be attributable to other time-invariant background variables that we have not assessed, such as differences in parenting practices, personality characteristics, or genetics [14].

The absence of significant associations in adjusted CLPMs for most of the study period, in combination with the high stability of educational level over time, highlights the importance of the transition to secondary school in educational systems characterised by early stratification. In line with past studies [15, 18], children's IQ and effortful control strongly predicted into which educational track participants were selected in early adolescence, which subsequently

predicted alcohol use in CLPMs. By determining initial educational level, these characteristics also predict subsequent trajectories in alcohol use. This explanation is further supported by the fact that we no longer find a significant association between IQ and early alcohol use once concurrent educational level is added as covariate. Furthermore, our findings highlight the importance of parental SES as determinant of both selection into educational tracks and early alcohol use. Studies from the Netherlands have consistently shown that children from lower SES households more frequently enter lower educational tracks, regardless of their performance on standardized tests [4]. Studies considering associations between parental SES and adolescent alcohol use have been less consistent [62–64], but a meta-analysis that specifically focused on early adolescence (age 10–15) found that youngsters from lower SES households consume more alcohol [64]. In our study, the association between parental SES and early alcohol use remains significant even after further adjusting for adolescents' educational level around age 14. This suggests that parental SES is an important determinant of alcohol use in early adolescence, above the effect of own educational level; perhaps because parental SES reflects differences in parental attachment, alcohol use, and alcohol-related permissiveness [19].

## Conclusions and implications

We mainly found evidence in support of the social causation theory in early adolescence, when lower education predicted increases in subsequent alcohol use. In young adulthood, we found tendencies towards opposite associations, with a stronger escalation of alcohol use amongst the higher educational tracks, though most of these effects failed to reach statistical significance in adjusted CLPMs. We found no evidence for health-related selection attributable to alcohol use throughout adolescence and young adulthood. The very high stability in educational level throughout adolescence might be typical of countries with educational systems characterized by an early selection and highlights the importance of determinants already present in childhood, which predict the initial selection into educational tracks. By determining educational level in early adolescence, these characteristics also predict subsequent inequalities in alcohol use.

Our findings emphasize the need for interventions to delay the early escalation of alcohol use amongst adolescents in the lower educational tracks. Early drinking is an important predictor of later problematic use and alcohol use disorders [65]. Background characteristics were not able to explain this early escalation, pointing towards educational differences in social norms and peer group composition in the educational context. Interventions may aim to integrate adolescents' social networks and popular peers in particular, who my act as key opinion leaders promoting good health behaviours [66, 67]. Peer-led interventions have been shown to reduce adolescent alcohol use [68]. While less in known about peer-led interventions for young adults, targeting social norms may also be effective in reducing drinking in young adults in college [69, 70].

## Supporting information

**S1 Fig. Path diagram of a bivariate cross-lagged panel model.**
(PDF)

**S2 Fig. Path diagram of a cross-lagged panel model adjusted for time-invariant baseline characteristics.**
(PDF)

**S3 Fig. Path diagrams of one-sided cross-lagged panel models with fixed effects according to the specification by Allison et al. [26]; separate fixed effects models were fit to assess**

each of the two hypothesized causal directions between educational level and alcohol use.
(PDF)

**S4 Fig. Bidirectional associations between educational level and alcohol use in the TRAILS Study (the Netherlands, 2000–2017, N = 2,229); sequentially adjusted linear regression coefficients (stdyx-standardized ß-coefficient, robust standard error, p-value) for different sets of covariates.** Model 1: bivariate cross-lagged panel model. Model 2: cross-lagged panel model adjusted for demographics (age, gender, area of residence, and ethnicity). Model 3: cross-lagged panel model adjusted for demographics, and parental socioeconomic status. Model 4: cross-lagged panel model adjusted for demographics, parental socioeconomic status, and adolescent psychological characteristics (IQ, effortful control). Edu = educational level; Alc = alcohol use. **Boldface** denotes statistical significance at p < 0.05.
(PDF)

**S5 Fig. Bidirectional associations between educational level and alcohol use in the TRAILS Study (the Netherlands, 2000–2017, N = 2,229); linear regression coefficients (stdyx-standardized ß-coefficient, robust standard error, p-value) from cross-lagged panel models without (Model 1 and 2) and with fixed effects (Model 3); the binge drinking item was removed from the AUDIT-C in these models.** Model 1: bivariate cross-lagged panel model. Model 2: cross-lagged panel model adjusted for age, gender, area of residence, ethnicity, parental socioeconomic status, IQ, and effortful control at baseline (wave 1). Model 3: cross-lagged panel models with fixed effects–adjustment for time-invariant characteristics was performed by inclusion of a latent variable. Edu = educational level; Alc = alcohol use. **Boldface** denotes statistical significance at p < 0.05.
(PDF)

**S6 Fig. Bidirectional associations between educational level and alcohol use in the TRAILS Study (the Netherlands, 2000–2017, N = 2,229); regression coefficients (stdyx-standardized ß-coefficient, robust standard error, p-value) from cross-lagged panel models without (Model 1 and 2) and with fixed effects (Model 3), using the Bayes estimator; educational level from wave 3 to wave 5 was declared as "categorical" in Mplus.** Model 1: bivariate cross-lagged panel model. Model 2: cross-lagged panel model adjusted for age, gender, area of residence, ethnicity, parental socioeconomic status, IQ, and effortful control at baseline (wave 1). Model 3: cross-lagged panel models with fixed effects–adjustment for time-invariant characteristics was performed by inclusion of a latent variable. Edu = educational level; Alc = alcohol use. **Boldface** denotes statistical significance at p < 0.05.
(PDF)

**S1 Table. Attrition analysis–characteristics of young adults remaining in the TRAILS Study (the Netherlands, 2000–2017, N = 2,229) at wave 6, compared to participants who had dropped out of the cohort between wave 2 and wave 5.** SD = standard deviation. P-values were computed using chi-squared tests for categorical variables and two-sample t-tests for continuous variables.
(PDF)

**S2 Table. Characteristics of participants with classifiable educational level compared to those with missing/unclassifiable educational level from wave 2 to wave 6 in the TRAILS Study (the Netherlands, 2000–2017, N = 2,229).** SD = standard deviation. [a]Alcohol use was measured using a quantity-frequency score from wave 2 to wave 5, and using the AUDIT-C at wave 6. P-values were computed using chi-squared tests for categorical variables and two-

sample t-tests for continuous variables.
(PDF)

**S3 Table. The association between baseline characteristics (wave 1) and alcohol use and educational level from wave 2 to wave 6 in the TRAILS Study (the Netherlands, 2000–2017, N = 2,229) in the multivariate-adjusted cross-lagged panel model (model 2) in Fig 2; linear regression coefficients (stdyx-standardized ß-coefficient, robust standard error, p-value); all predictors are mutually adjusted.** All predictors are mutually adjusted. From wave 3 to wave 6, all predictors are additionally adjusted for educational level and alcohol use in the preceding wave. **Boldface** denotes statistical significance at p < 0.05.
(PDF)

**S4 Table. Post-hoc analysis–the association between baseline characteristics (wave 1) and alcohol use at wave 2 in the TRAILS Study (the Netherlands, 2000–2017, N = 2,229) in the multivariate-adjusted cross-lagged panel model (Fig 2, Model 2) after additionally regressing wave 2 alcohol use on wave 2 education; linear regression coefficients (stdyx-standardized ß-coefficient, robust standard error, p-value); all predictors are mutually adjusted.** All predictors are mutually adjusted. **Boldface** denotes statistical significance at p < 0.05.
(PDF)

## Acknowledgments

The authors would like to thank the participants of the TRAILS Study and all staff involved in the management and execution of this project.

## Author Contributions

**Conceptualization:** Heiko Schmengler, Margot Peeters, Anton E. Kunst, Albertine J. Oldehinkel, Wilma A. M. Vollebergh.

**Data curation:** Heiko Schmengler.

**Formal analysis:** Heiko Schmengler.

**Funding acquisition:** Margot Peeters, Anton E. Kunst, Albertine J. Oldehinkel, Wilma A. M. Vollebergh.

**Investigation:** Heiko Schmengler.

**Methodology:** Heiko Schmengler, Margot Peeters, Wilma A. M. Vollebergh.

**Project administration:** Heiko Schmengler, Margot Peeters, Wilma A. M. Vollebergh.

**Supervision:** Margot Peeters, Anton E. Kunst, Albertine J. Oldehinkel, Wilma A. M. Vollebergh.

**Visualization:** Heiko Schmengler.

**Writing – original draft:** Heiko Schmengler.

**Writing – review & editing:** Heiko Schmengler, Margot Peeters, Anton E. Kunst, Albertine J. Oldehinkel, Wilma A. M. Vollebergh.

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
