## [Decision Letter · Decision Letter 0]

26 Jul 2021

PONE-D-21-07212

Educational level and alcohol use in adolescence and early adulthood – The role of social causation and health-related selection – The TRAILS Study

PLOS ONE

Dear Dr. Schmengler,

Thank you for submitting your manuscript to PLOS ONE. After careful consideration, we feel that it has merit but does not fully meet PLOS ONE’s publication criteria as it currently stands. Therefore, we invite you to submit a revised version of the manuscript that addresses the points raised during the review process.

We look forward to receiving your revised manuscript.

Kind regards,

Matthew J. Gullo

Academic Editor

PLOS ONE

Journal Requirements:

This study is made possible by the Consortium on Individual Development (CID). CID is funded through the Gravitation program of the Dutch Ministry of Education, Culture, and Science and the Netherlands Organization for Scientific Research (NWO) (grant number 024.001.003). This research is part of the TRacking Adolescents’ Individual Lives Survey (TRAILS). Organizations participating in TRAILS include various departments of the University Medical Center and University of Groningen, the Erasmus University Medical Center Rotterdam, Utrecht University, the Radboud Medical Center Nijmegen, and the Parnassia Bavo group, all in the Netherlands. TRAILS has been financially supported by various grants from NWO, ZonMW, GB-MaGW, the Dutch Ministry of Justice, the European Science Foundation, BBMRI-NL, and the participating universities. The authors would like to thank the participants of the TRAILS Study and all staff involved in the management and execution of this project.

TRAILS has been financially supported by various grants from NWO, ZonMW, GB-MaGW, the Dutch Ministry of Justice, the European Science Foundation, BBMRI-NL, and the participating universities.

Reviewers' comments:

Reviewer's Responses to Questions

**Comments to the Author**

1. Is the manuscript technically sound, and do the data support the conclusions?

Reviewer #1: Partly

Reviewer #2: Partly

2. Has the statistical analysis been performed appropriately and rigorously? 

Reviewer #1: Yes

Reviewer #2: Yes

3. Have the authors made all data underlying the findings in their manuscript fully available?

Reviewer #1: No

Reviewer #2: No

4. Is the manuscript presented in an intelligible fashion and written in standard English?

Reviewer #1: Yes

Reviewer #2: Yes

5. Review Comments to the Author

Reviewer #1: Abstract

• You may want to include the number of waves in the methods section

Introduction/ Literature Review

• Generally, this section is written well and substantiates the issue the authors want to examine.

• You may want to be more explicit about your research question in the current study section. You could also incorporate the theories you are leveraging if you wanted to add the direction of effects into your research questions.

Methods

• Can you include the omega reliability for EATQ-R and AUDIT-C rather than alpha? Also, for the measures, can you include the number of items? Also, binge drinking is defined differently for males and females, was this adjusted for in that item? It looks like six or more drinks were used for both males and females.

• The methods section is clear. I appreciate the detailed analytic approach. Why wasn’t the CLPM with fixed effects used as the primary model (and remove the CLPM)? The authors cite Hamaker’s work, so they are familiar with the issue of the traditional CLPM (convergence effect aka “smushed effect.” Presenting Allison’s approach or the RI-CLPM (or the ALT-SR model see Curran) as the primary model would be more appropriate, mainly since this is a longitudinal study. Partitioning variance at the within- and between-person levels seems more suited to examine changes across these developmental stages. You could also conduct a Chi Bar test to test whether a model like the RI-CLPM would be warranted. I think the RI-CLPM is a good approach because you can report both between-person intercept (across time or at each time point) as well as within-person lagged effects. A focus on this approach would provide more information, examine lagged effects at a more appropriate level of examining developmental changes, and addressing the significant issues and limitations of the CLPM (see Berry and Willoughby, 2017)

• I’m also wondering if you could present the Intra Class Correlations so readers could get a sense of how much variance in the variables are within- or between people. I was most interested in the ICC for educational level. Is there wave-to-wave fluctuation in education level, or do most people remain on the same track?

• The authors state, “male gender, non-Dutch ethnicity lower educational level, IQ, and effortful control, as well as those from lower SES households, were more likely to drop out of the study.” Since FIML was used to account for missing data, you could also note that these variables were also included in the model, thereby adjusting for potential bias due to missing data on these variables. Also, you may want to report what the actual attrition is by showing the sample size at each wave since the table is only reported in supplemental.

• Are you using standard cutoffs for the fit indices listed to assess model fit? For example, are you following Hu and Bentler, 1999? I understand there are issues with cutoffs, but sometimes it’s good to state what you are using to guide the model fit assessment.

• I think the most crucial point is how each of the variables was treated in the CLPM. I could see one arguing that alcohol use was treated as a continuous variable; however, treating educational level as a continuous variable is not appropriate. I suggest you define this variable as categorical in Mplus and use the WLSMV estimator with theta parameterization, which is a probit model. You could also choose to use marginal maximum likelihood with a logit or probit link so you could invoke FIML to address missing data; however, remember this approach does not have a baseline model and thus has no fit indices. You could use this as a sensitivity analysis for the WLSMV model. The primary point is that it is not appropriate to treat an ordered categorical variable as continuous as it appears been done in the current analysis. This holds true for the linear regression models and the CLPM.

• Can you add the age, gender, area of residence, and ethnicity at baseline in Table 3? And create a table for them for the CLPM?

Results

• You may want to try to incorporate your research questions in the result section. You could also include social causation and health selection in this section.

• In general, the results are organized well. You could incorporate standardized effects in the results section so readers can get a sense of the magnitude of the effects.

• Table 2: I do not think a test like ANOVAs should be conducted for descriptive tables. There are so many tests you had to run (10 comparisons for each variable across time). Were there any adjustments made to account for all these tests? Also, is it possible to clarify the differences in the subscript (a, b, c)?

• Table 3: This linear regression model seems redundant. These effects are included in Model 2 of the CLPM, and it is more appropriate to report these findings rather than fitting another model to report demographic predictors. Also, why aren’t the controls include in the table? Further, Model 3 Table 3 note states that SES is included, but no effect is presented.

• Within time correlations (residual correlations) are not in the figure. Can you add them to a table? Maybe supplemental?

• What do you make of the difference between the CLPM and the Fixed effects model? I would argue that it is essential to plausibly disaggregate within- and between-person effects. In my view, all longitudinal models are inherently multilevel because persons are context. As such, it is important to disaggregate these substantively different levels of variance. How one differs from their own mean or trajectory over time (within-person) is not the same thing as how one differs from someone else (between-person). Both are meaningful but carry different substantive meanings. Allison’s approach is good; however, in the current study, the between-person effects should be interpreted as well (they are meaningful).

Discussion

• Other limitations include the heavy use of single-item indicators, which assume measurement invariance over time. Further, not addressing the issue of polysubstance use, which is very common. As such, any harm attributed to alcohol may be better explained by the use of multiple substances.

• I appreciate the discussion of the findings. Can you add more about the practical and policy implications of the current work?

Other

• I suggest that you temper the use of the word causation. The data and models are correlational. In addition, several confounds were not included in these models, and the study does not address the issue of polysubstance use, which has shown to be very common across this age range. As such, it is unknown in this study if harms associated with alcohol use are being misappropriated.

• Is this project preregistered on OSF? If so, you could include the link.

• Were the scripts uploaded to GitHub? If so, could you make the repository public and share the link?

• It appears that a lot of post hoc analyses were run in this study. I encourage the authors to focus on the primary models and limit the number of fitted models. The more models you run, the higher the likelihood of capitalizing on chance. Remember, at a p-value of .05, one out of 20 models is wrong, given chance alone.

Reviewer #2: The manuscript reports on a study investigating reciprocal associations between alcohol use and educational attainment. It benefits from a large sample and advanced analytic approach that controls for key covariates.

1) The study does not comply with the PLOS data policy: “Why do we not allow an author to be the only point of contact for fielding requests for access to restricted data?

When possible, we recommend authors deposit restricted data to a repository that allows for controlled data access. If this is not possible, directing data requests to a non-author institutional point of contact, such as a data access or ethics committee, helps guarantee long term stability and availability of data. Providing interested researchers with a durable point of contact ensures data will be accessible even if an author changes email addresses, institutions, or becomes unavailable to answer requests.”

2) “Social causation” and “health-related selection” should both be defined in the first paragraph of the abstract.

3) Page two, line 62: Please remove reference to “causation” as the data is not from an experiment incorporating randomization. Please use “social causation theory” or “social causation hypothesis” throughout the manuscript to avoid confusion.

4) Page 3, line 78: it is more correct to say that youngsters can affect their own SES through education.

5) No psychometric evidence is provided for the validity of the quantity-frequency measure. How well did it correlate with audit-c? Can the authors provide any other evidence that the measure is reliable and valid?

6) Has the method used to classify adolescent educational level been used in a published study before? Is it considered a valid way to quantify educational level?

7) I have not come across the approach taken for the analogous CLPM with fixed effects models. Can the authors provide another reference, aside from Allison, that has used this approach to give confidence in the validity of the approach? It is not clear what it adds over the analytic approaches.

8) The data were not missing completely at random (MCAR). While Full Information Maximum Likelihood (FIML) estimation is appropriate to account for this, it should be clear that this is only case for models that include auxiliary variables predictive of missingness, i.e. those that include gender, ethnicity, and IQ. Differences between these models (i.e., Model 3) and others may be due to bias in accounting for missing data.

9) The linear regressions reported in table 3 are not well justified. What exactly are they trying to demonstrate? If it was predictors of educational track, wasn’t this already determined at wave 1? These regressions are also prone to bias related to missing data. Unlike FIML with auxiliary variables, this bias is not adequately accounted for. I think these regressions should probably be removed unless a compelling rationale can be provided.

6. PLOS authors have the option to publish the peer review history of their article (what does this mean?). If published, this will include your full peer review and any attached files.

Reviewer #1: **Yes: **Gabriel J Merrin

Reviewer #2: No

---

## [Author Response · Author response to Decision Letter 0]

2 Oct 2021

Please see the file response_to_reviewers.docx

---

## [Decision Letter · Decision Letter 1]

9 Nov 2021

PONE-D-21-07212R1Educational level and alcohol use in adolescence and early adulthood – The role of social causation and health-related selection – The TRAILS StudyPLOS ONE

Dear Dr. Schmengler,

Thank you for submitting your manuscript to PLOS ONE. After careful consideration, we feel that it has merit but does not fully meet PLOS ONE’s publication criteria as it currently stands. Therefore, we invite you to submit a revised version of the manuscript that addresses the points raised during the review process.

We look forward to receiving your revised manuscript.

Kind regards,

Matthew J. Gullo

Academic Editor

PLOS ONE

Journal Requirements:

Reviewers' comments:

Reviewer's Responses to Questions

**Comments to the Author**

1. If the authors have adequately addressed your comments raised in a previous round of review and you feel that this manuscript is now acceptable for publication, you may indicate that here to bypass the “Comments to the Author” section, enter your conflict of interest statement in the “Confidential to Editor” section, and submit your "Accept" recommendation.

Reviewer #1: All comments have been addressed

Reviewer #2: (No Response)

2. Is the manuscript technically sound, and do the data support the conclusions?

Reviewer #1: Yes

Reviewer #2: Yes

3. Has the statistical analysis been performed appropriately and rigorously? 

Reviewer #1: Yes

Reviewer #2: Yes

4. Have the authors made all data underlying the findings in their manuscript fully available?

Reviewer #1: Yes

Reviewer #2: No

5. Is the manuscript presented in an intelligible fashion and written in standard English?

Reviewer #1: Yes

Reviewer #2: Yes

6. Review Comments to the Author

Reviewer #1: Introduction/Literature Review

• The authors may want to also mention additional limitations to the CLPM, namely the convergence effect such that the constructs combine both within and between-person variance.

• I think the introduction and literature review are organized well and substantiate the issue addressed in the manuscript.

Methods

• Can you include omega reliability rather than alpha?

• The methods are straightforward and organized

• There were significant associations with missing participants and some of the variables. You may want to include a sentence that states you included all these variables in your model to adjust for potential bias due to missing data on these variables.

• I like the inclusion of the sensitivity analyses. You may consider including them in supplemental

Results

• The results are organized well. You may consider adding a few subheadings.

• Do you think the timing of each wave may have impacted the findings? For example, so are spaced further apart in time.

Discussion

• I appreciate the findings were placed in context and discussed in terms of practical implications.

Reviewer #2: The authors have addressed most of my concerns. One final comment relates to the statement on p8: line 201: It is more in line with the cited evidence for this to be written as, "Quantity-frequency measures of alcohol use have shown adequate/good validity and reliability across studies [30]."

7. PLOS authors have the option to publish the peer review history of their article (what does this mean?). If published, this will include your full peer review and any attached files.

Reviewer #1: No

Reviewer #2: No

---

## [Author Response · Author response to Decision Letter 1]

23 Nov 2021

Please see the file review2_response form.docx.

---

## [Editor Report · Decision Letter 2]

7 Dec 2021

Educational level and alcohol use in adolescence and early adulthood – The role of social causation and health-related selection – The TRAILS Study

PONE-D-21-07212R2

Dear Dr. Schmengler,

We’re pleased to inform you that your manuscript has been judged scientifically suitable for publication and will be formally accepted for publication once it meets all outstanding technical requirements.

Kind regards,

Matthew J. Gullo

Academic Editor

PLOS ONE
---

## [Editor Report · Acceptance letter]

14 Dec 2021

PONE-D-21-07212R2 

Educational level and alcohol use in adolescence and early adulthood – The role of social causation and health-related selection – The TRAILS Study 

Dear Dr. Schmengler:

I'm pleased to inform you that your manuscript has been deemed suitable for publication in PLOS ONE. Congratulations! Your manuscript is now with our production department. 

Kind regards, 

on behalf of

Assoc. Prof. Matthew J. Gullo 

Academic Editor

PLOS ONE